# Structural, Thermal, and Optical Studies of Gamma Irradiated Polyvinyl Alcohol-, Lignosulfonate-, and Palladium Nanocomposite Film

**DOI:** 10.3390/polym14132613

**Published:** 2022-06-28

**Authors:** Foued Gharbi, Kaoutar Benthami, Tarfa. H. Alsheddi, Mai M. E. Barakat, Nisrin Alnaim, Adil Alshoaibi, Samir A. Nouh

**Affiliations:** 1Department of Physics, College of Science, King Faisal University, P.O. Box 400, Al-Ahsa 31982, Saudi Arabia; talsheddi@kfu.edu.sa (T.H.A.); nalnaim@kfu.edu.sa (N.A.); aalshoaibi@kfu.edu.sa (A.A.); 2Unité de Radioanalyse, Centre National des Sciences et Technologies Nucléaires, Technopôle de Sidi Thabet, Sidi Thabet 2020, Tunisia; 3Physics Department, Facult’e des Sciences, Universit´e Moulay Ismail, B.P. 11201, Meknes 50050, Morocco; benkaoutar@yahoo.fr; 4Physics Department, Faculty of Science Yanbu, Taibah University, Yanbu 41912, Saudi Arabia; maibarakat_phy@yahoo.com; 5Physics Department, Faculty of Science, Alexandria University, Alexandria 21500, Egypt; 6Physics Department, Faculty of Science, Taibah University, Medina 44256, Saudi Arabia; snouh@taibahu.edu.sa; 7Physics Department, Faculty of Science, Ain Shams University, Cairo 11865, Egypt

**Keywords:** nanocomposites, structure, thermal analysis, UV spectroscopy, radiation

## Abstract

Nanocomposite (NC) films of polyvinyl alcohol (PVA), lignosulfonate (Lg), and nanosized palladium (Pd) were synthesized by ex-situ casting method. Samples from the synthesized PVA-Lg/Pd NC films were irradiated with 5–100 kGy γ doses. The effect of γ doses on the structural, thermal, and optical characteristics of the NC films were studied using different characterization techniques. The results indicated that the γ irradiation improves the decomposition temperature from 227 to 239 °C, signifying an increase in the thermal stability of the NC films. This was accompanied by a reduction of the melting temperature due to the increase of the amorphous phase. This can be attributed to the dominance of crosslinking. On the other hand, the refractive index increased from 2.21 to 2.32 while increasing the γ dose up to 100 kGy. This was associated with a reduction of the optical bandgap from 3.49 to 3.30 eV, which could be attributed to the increase in the amorphous phase as a result of crosslinking. This indicates an enhancement of the spreading of the NPs inside the blend matrix due to γ irradiation. This results in a more compacted construction of the PVA-Lg/Pd NC films. Furthermore, we used the Commission Internationale de E’Claire (CIE) method to estimate the change in color among the irradiated NC films and the pristine film. The PVA-Lg/Pd NC attained a significant color difference value greater than five, meaning permanent color changes.

## 1. Introduction

Nowadays, the fabrication of nanocomposites (NCs) using polymer blends and nanoparticles (NPs) has attained the consideration of many authors owing to the possibility of its use in several fields [1]. These polymeric NCs have an individual character, unlike those of bulk, due to the tiny size of the embedded NPs [2]. Thus, those NCs can be widely used is many fields such as advanced coatings, single electron transistors, sensors, optoelectronic devices, and solar cells [3].

Polyvinyl alcohol (PVA) has an excellent film-developing ability and excellent transparency. It is one of the most representative host polymers to NPs due to its significant chemical resistance, optical characteristics, and bio-compatibility [4]. The composites manufactured from PVA can be used in several applications including coatings [5], optical membranes [6], nano-fibers [7], and wrapping matter [8].

The main building units of lignosulfonate (Lg) biopolymer are guaiacyl and p-hydroxyphenyl which are randomly connected through C−C or ether bonds [9]. Lg is an abundant natural phenol polymer which can be used in several applications concerning non-wooded and wooded biomasses [10]. Lg is a preferred compatible matter for polymer blends. For example, tiny micrometers of Lg can interact strongly with PVA through H_2_ bonding [11]. An earlier study was carried out to investigate whether crosslinking predominates when Lg interacts with PVA [12]. This can be attributed to the abundant groups in Lg, such as the methoxyl groups, H_aro_ and the OH group [13]. Lignosulfonate is suitable for blending with manufactured polymers to improve their properties [14]. It is a sensible additive to polymeric matters owing to its great thermostability and radical capture ability [15].

Polymer NCs have drawn great attention due to their enhanced physical and chemical properties. The embedding of NPs within the polymer matrix can produce high quality films with better selectivity and sensitivity [16]. The inclusion of metallic NPs within the polymeric matrix enhances its physical character and thus the resultant NC will be reasonable for several applications [17,18,19]. Palladium NPs initiate application in bio-sensing, *C*–*C* coupling reactions, and surface-enhanced Raman spectroscopy [20]. The inclusion of NPs treats the disadvantages that arise from the narrow absorption bands and the degradation of the polymeric matter [21,22]. This can be achieved by developing their size-dependent spectral tunability and intrinsic material stability [22].

The number of amorphous regions in polymers has an important role in characterizing them. Besides, thermogravimetric analysis (TGA) is extensively applied to investigate the thermostability of polymers by providing information concerning the kinetic parameters of the thermal decomposition character.

Some polymers have several significant characters that gave them huge technological and financial importance. However, in spite of this, they have some problems associated with changes in their physical character. One problem, for example, is the poor thermostability owed to the structural deficiencies that were formed during the step of polymerization. Thus, it is essential to stabilize these polymers using γ radiation [23].

The γ radiation causes breakage of the polymer chains, thus creating free radicals that are chemically active. These active free radicals cause the formation of new bonds among the chains via crosslinking. This affects the macro molecular structure and morphology of the polymeric NCs [23]. Additionally, the optical character of materials represents an important criterion for researchers owing to the extensive applications in photo electronic instruments [24]. Also, the change of color in polymers due to γ irradiation is an essential property which aids in interpreting the modification in the polymer properties. It can be applied in marketable applications, including radiation processing and dosimetry [25].

In the present study, we synthesized a polymer blend from polyvinyl alcohol and lignosulfonate. The resultant blend was used as a host material for palladium nanoparticles aiming to obtain a novel nanocomposite of enhanced thermal and optical properties. Finally, the synthesized nanocomposite was irradiated with γ radiation with the aim of investigating the possibility of improving its properties to be used in industry.

## 2. Experimental

### 2.1. Materials

We purchased the sodium Lg (∼94%, SLS and Mn = 7000) from Sigma–Aldrich Company, St. Louis, MO, USA, while we obtained the PVA from Sigma–Aldrich GmbH, Cairo, Egypt. Polyethylene glycol, methanol HPLC-grade, palladium chloride, methylene chloride, hexane, and acetone were obtained from Merck, Kenilworth, NJ, USA.

The Pd NPs were synthesized and characterized following the same technique used in our previous study [26]. The Pd has a particle size in the range 2–22 nm with 10 nm on average [26].

The PVA-Lg/Pd NC was prepared by means of casting technique. We dissolved pure Lg and PVA (50/50 w%) in 100 mL of hot deionized water through strong stirring for 4 h at 85 °C. Then, we added 0.5 w% of the formerly prepared NPs to the PVA-Lg solution (2 g in 30 mL), through magnetic stirring following the equation:(1)$$x\left(\mathrm{wt}\%\right)=\frac{w_{f}}{w_{p}+w_{f}}\times 100$$
where w_p_ is the weight of PVA-Lg blend and w_f_ is the weight of Pd NPs. Then the mixtures were cast into Petri dishes and dried in a vacuumed oven at 80 °C. Finally, we fixed the attained films on a plate at 40 °C for 96 h to remove the remaining solvents. The film thickness (0.15 mm) was estimated using a thickness gauge (Model 11/2704 Ast MD 370 standard) that was calibrated by the Arab British Dynamics (Cairo, Egypt).

### 2.2. Irradiation Facility

A ^60^Co source (Canada A.E.A Ltd., Ottawa, ON, Canada) of dose rate 1 kGy/h was used. The irradiation was performed at the NCRRT of Egyptian Atomic Energy Authority, Cairo, Egypt.

### 2.3. Experimental Apparatus

X-ray diffraction (XRD) was carried out by Shimadzu 6000. The diffractometer was operating with Cu-kα ray of wavelength of 1.5406 A° and scanned in the 2θ range 10–60°, at a 2°/min speed.

Fourier transform infrared spectroscopy (FTIR) was achieved, in the wavenumber range 400–4000 cm^−1^, with a Shimadzu spectrophotometer (Type 8201 PC, with precision ± 4 cm^−1^).

The thermal measurements were conducted using a Shimadzu-50 instrument (Shimadzu, Tokyo, Japan). The TGA curves were measured in the temperature range from room temperature up to 500 °C at 10 °C/min. For differential thermal analysis (DTA) scans, we used α-Al_2_O_3_ powder as a reference matter. The scans were obtained in the temperature range from room temperature up to 300 °C, at 10 °C/min, with nitrogen gas flow rate 20 cm^3^/min.

The absorbance records, in the wavelength from 200 up to 800 nm, were collected using a Shimadzu UV spectrophotometer, Ttype 3101 PC, Berkshire, UK. The Commission Internationale de E’Claire (CIE) was used to determine any color variation between the irradiated samples and the pristine sample. All the mathematical equations used were presented in detail in our previous publication [27].

## 3. Results and Discussion

### 3.1. Structural Investigation

#### 3.1.1. XRD

The XRD study was carried out at the range of 2θ (10–60°) and results are displayed in Figure 1. The patterns of the NC films showed the semi-crystalline nature of the NC having a major amorphous phase. A wide diffraction peak appeared at 2θ = 18.5° that almost matched the (101) reflection plane of the PVA polymer [28].

The variations in the diffraction pattern were predictable owing to the injection of NPs; nevertheless, we did not observe any diffraction peak belonging to the NPs in the pattern of the NC film, signifying a full dispersion of the Pd NPs in the PVA-Lg matrix. The integral intensity (I) of the broad diffraction peak, which refers to the area under the peak, was calculated and characterized considering the γ dose in Figure 2. The values of I increased with the increasing dose up to 10 kGy, then decreased with higher doses up to 100 kGy. We attribute the increase in I to degradation that causes a growth in the amount of the ordered regions in the NC and reduces the intermolecular stress in the amorphous regions. This enhances the mobility of chains and allows macromolecules to be reordered [29]. Comparatively, the γ doses in the range 10–100 kGy damage the crystalline portions and change the ordered arranged areas into irregular ones by creating hydrogen bonds among the NPs and the blend chains due to crosslinking. Since crosslinking enhances the amorphous phase in the NC film, then the NC films may be appropriate candidates for several industrial requests that require bending without contravention.

The width of peak at half of the maximal intensity (ΔW) is related to the crystallite size (L). Thus, we used the Scherrer equation to estimate the values of ΔW: (2)L=(0.89λ)/(ΔWCosθ) 
where λ is the wavelength of the X-rays. The change of ΔW with γ dose is displayed in Figure 2. There is no significant change in ΔW that means no change in the width of the lamella.

#### 3.1.2. FTIR

In order to illustrate the structural modification in the NC films due to γ irradiation, FTIR spectroscopy was conducted. The induced modifications were evaluated considering the variation in peak intensity that fits each function group. Figure 3 shows the FTIR spectra of the irradiated NC samples and the pristine sample.

The functional groups relating to PVA should appear at 851, 1138, 1433, 2940, and 3360 cm^−1^ and match the C–C–O stretch (alcohol), CH_2_ deformation (twist or wag), CH_2_ deformation coupled with O–H deformation (alcohol), C–H stretch (CH_2_ asymmetric stretch), and O–H stretch (alcohol), respectively [30]. The functional groups of the PVA-Lg/Pd NC were observed at 860, 1100, 1430, 2925, and 3350 cm^−1^, correspondingly. When we compare the values of wavenumbers of the pure PVA with those of the NC, we observe that the peaks of the PVA-Lg/Pd NC are slightly shifted to higher or lower wavenumbers. We attribute this to the damage in the C–H bond due to the impeding of the Pd NPs which aids the creation of carbon loaded structure and hydrogen free radicals [31].

The bands representing the C–O stretching vibrations are due to the residual acetate groups next the manufacture of PVA from oxidation or hydrolysis of polyvinyl acetate during its manufacture. Also, the C–C bands may be due to crosslinks realized while the polymer was heated during the NC preparation [32,33]. These bands were decreased with doses up to 25 kGy and then increased while raising the dose up to 100 kGy. The band representing the C–C–O stretching is predictable to the crystallization [34] and signifying the semi-ordered character of PVA. This band exhibited a non-monotonic trend with the γ dose. Moreover, the O–H stretching band is basically characteristic of phenols and alcohols. This band nearly decreased when the dose reached 100 kGy, meaning there is damage to crystalline structure, confirming the predominance of crosslinking. The CH_2_ bending vibration, the CH deformation vibration (1330 cm^−1^), and the CH_2_ asymmetric stretching bands showed identical trend as they reduced while raising the γ dose up to 100 kGy. We attribute the reduction in their intensity to the ionizing effect of γ radiation that damages the C–H bond creating free radicals that react with OH groups causing crosslinking. Generally, the change in band intensity can not only be attributed to the ionizing effect of γ photons but also to the replacement of some of the carbon atoms by Pd NPs in the backbone of the polymer blend matrix.

### 3.2. Thermal Investigation

#### 3.2.1. TGA

TGA was carried out on the PVA-Lg/Pd NC films to obtain information about the changes in its thermal stability with the γ dose. TGA was applied at a heating rate of 10 °C/min and at the temperature range from RT up to 500 °C. The TGA curves for the γ irradiated NC samples and pristine sample are shown in Figure 4. The decomposition of NC samples occurred in two steps. The degradation temperatures of the two weight loss steps (T_o_, T_1_) could be evaluated from the TGA curves and are displayed in Table 1. The numerical values of T_o_ and T_1_ decreased while increasing the dose up to 10 kGy, then increased while raising the γ dose up to 100 kGy. We attribute this trend to the initial scission, followed by the dominance of crosslinking that enhances the thermal stability of the NC.

Calculation of the activation energy of thermal decomposition, E_a_, is convenient for investigating the thermal stability of the NC films. Numerous thermo-gravimetric procedures use the heating rate to evaluate E_a_. The method presented by Horowitz and Metzger [35] was used in the present study. The values of E_a1_ and Ea_2_ for the two decomposition steps are displayed in Table 1. The activation energies for the two weight loss steps exhibited a similar character to that of the T_o_, signifying extra thermal stability with a reduced rate of decomposition when increasing γ dose in the range from 10 to 100 kGy. The γ irradiation with doses up to 10 kGy causes the degradation of the CH and CH_2_ bonds that formed chemically active free radicals [36,37]. An additional factor that contributes to the breaking of bonds is thermal degradation. Extended heating breaks the bonds randomly and creates steady molecules with fewer molecular weights. Additionally, heat splits the small molecular products due to the reaction of side groups without changing the initial molecular weight. In other words, the produced free radicals create chemical reactions leading to the development of novel bonding via crosslinking, disturbing the chemical construction of the NC and thereby improving its thermal stability.

#### 3.2.2. DTA

To evaluate the transition temperatures of the pristine and irradiated PVA-Lg/Pd NC films, DTA was carried out. The measurements were applied from room temperature up to 300 °C at 10 °C/min heating rate. The thermograms of the NC films are displayed in Figure 5. The DTA curves showed an endothermic peak due to melting (T_m_). The melting temperature appeared as a range of non-definite temperatures. This is due to the variety in chain length and the degree of freedom of the polymeric chains [38]. The numerical values of T_m_ were estimated and are displayed in Table 1. The values of T_m_ increased while raising the γ dose up to 10 kGy due to degradation, then decreased while raising the dose up to 100 kGy due to crosslinking which damaged the crystalline structure.

The mobility of the short chains is great, thus permitting the reorientation of molecules and form ordered segments. This resembles the cage effect that contains the free radical recombination before sharing in interactions that motivate crosslinking [39,40]. Consequently, the observed difference between the variation of degradation temperatures and the melting temperature of the dose is due to the fact that T_m_ identifies the crystalline regions of the polymeric NC films. It is likely that the low γ doses increases the thickness of ordered constructions. Comparatively, the high γ doses (10–100 kG) produce defects that split the crystals reducing T_m_ [41].

### 3.3. Optical Investigation

#### 3.3.1. Absorption Coefficient Investigation

The UV absorbance (*A*) of the irradiated samples and the pristine sample was measured. We then used the absorbance data to calculate the absorption coefficient (*α*) that describes the amount of light absorbed by a given thickness of a matter. We calculated the absorption coefficient applying the equation:(3)$$\alpha =\frac{\mathrm{Log}\;A}{d}$$
where *d* is the thickness of the sample.

The spectra of the absorption coefficient of the pristine and γ irradiated PVA-Lg/Pd NC films are displayed in Figure 6. The absorption coefficient increased while increasing the energy. Certain authors attributed the increase in absorbance with energy to the π−π* phenyl ring and locally excited transition n−π* among the energy levels [42]. This was related to the amount of conjugation between neighboring phenyl rings in the PVA chains [43]. Other authors attributed it to the creation of color centers [44]. This indicates that the occurrence of photochemical reactions in the NC matrix owes to the absorption of UV light that activates the macro-molecules to its single or triple state [42]. Additionally, the absorption coefficient increased with the γ dose, meaning that the intermediate atoms of the NC have absorbed the incident photon energy. This can be attributed to the Rayleigh scattering after the embedded Pd NPs came together with the induced modifications in the energy levels due to irradiation [45]. The energy transferred by the incident γ photons creates novel chemical configuration enhancing the absorbance. Also, the presence of more negative charges creates novel interior bonds in the NC. Besides, the minute size of the Pd NPs decreases the atomic volume occupied by Pd and thus increases its density, leading to the enhancement of the absorbance [46,47].

#### 3.3.2. Urbach Energy Investigation

The disordered materials have tail states in the gap area below the main absorption edge [48]; these can be calculated from the absorption coefficient (*α*) following the Urbach rule [49];
(4)$$\alpha ={\;\alpha }_{o}\;\exp\;\left(\frac{h\nu }{E_{U}}\right)$$
in which, *α*_o_ is a constant that defines the matter and E_u_ is the Urbach energy, which refers to the width of the tail of localized states in the forbidden bandgap [50]. The values of E_u_ were obtained from the slope of exponential dependence of the absorption coefficient edge vs. energy (Figure 6). The dependence of E_u_ on the γ dose is displayed in Figure 7. The numerical values of E_u_ increased from 0.30 to 0.61 eV while raising the dose up to 100 kGy. This is correlated with the development of the disordered phase caused by γ irradiation [51].

#### 3.3.3. Band Gap Investigation

The values of bandgap (Eg) were estimated using the following principle [52];
(5)$$\alpha h\nu =B{\left(h\nu -E_{g}\right)}^{n}\;\mathrm{true}\;\mathrm{for}\;E\;>\;E_{g}$$
in which B is a constant, hν is the energy of the photon, and n is an index of a value signifying the type of transition [53]. The E_g_ values of amorphous materials could be calculated by drawing (*α*hν)^0.5^ as a function of hν and then prolonging the linear section of the curve to the hν axis. The Tauc plots used to obtain energy gaps are presented in Figure 7.

The change of both E_u_ and E_g_ of the NC films with the γ dose is displayed in Figure 8. The values of E_g_ exhibited an opposite trend to those of E_u_, where they decreased while raising the dose up to 100 kGy. We attribute the decrease of E_g_ to the increase of the amorphous regions in the NC films. This encourages the localized states and generation of defects inside the E_g_ arrangement, initiating microelectronic transitions of lower energy. Furthermore, the significant outcomes of γ radiation on the NC samples were the generation of chemical active free radicals via the chain scissions that causes the development of conjugated bonds, decreasing the E_g_ [54].

#### 3.3.4. Refractive Investigation

A significant factor used in several optoelectronic applications is the real part of the complex refractive index (n). The values of n were estimated using the equation:(6)$$n=\left(\frac{1+R}{1-R}\right)+\sqrt{\frac{4R}{{\left(1-R\right)}^{2}}-k^{2}}$$
in which R is the reflectance which was estimated from the absorption spectrum following $R=1-\sqrt{{\mathrm{Te}}^{A}}$ (T is the transmittance and k is the extinction coefficient). K was estimated using the formula:(7)k=(λα/4π)

The values of n were plotted against the γ dose in Figure 9a. The refractive index increased while raising the dose up to 100 kGy. We attribute this trend to crosslinking, following the interpretation introduced by Shams-Eldin et al. [55] and Ranby & Rebek [56], where they attributed the decrease of n to degradation, while the increase of n could be due to the crosslinking of chains.

Since the dielectric parameters provide knowledge about the optical properties of matter [57], the dielectric constant (εʹ) was calculated using the values of k and by applying the formula [58]:(8)$$\varepsilon ʹ=n^{2}-k^{2}$$

Figure 9b shows the change of εʹ with hν. The variation of εʹ with hυ indicates the reactions between photons and electrons in the NC within this range of energies. The εʹ values increased while raising the dose up to 100 kGy, signifying an improvement of the density of states in the forbidden gap [59].

### 3.4. Color Difference Investigation

The estimation of color changes is an essential process in the field of radiation dosimetry. The red, green, and blue colors are symbolized by the scientific sets, X, Y, and Z, respectively, known as the tristimulus values [60]. Also, color saturation is symbolized by x, y and z, known as the chromaticity coordinates. Both (X, Y, Z) and (x, y, z) were computed using the method which we used in our previous work [27] and results are displayed in Table 2. In applying this method, we used the transmission values (370–780) calculated from the absorbance values represented in Figure 6. The values of X, Y, and Z decreased while raising the dose up to 100 kGy. An increase in the values of x and y with the γ dose was noticed. Conversely, the values of z decreased. 

Following the CIELAB system, the color intercepts are aces symbolized by a*, b*, and L*. The intercept a* expresses the red (+a*) and green (−a*) components, while b* expresses the yellow (+b*) and blue (−b*). L* expresses the lightness. The white has an L* of 100, and the black has an L* of 0. The precision in computing L* is ±0.05 and ±0.01 for both a* and b*, correspondingly. The change in a*, b*, and L* with the γ dose is displayed in Figure 10a. The color intercepts a* and b* showed negative values that increased while raising the dose up 100 kGy. This signifies that the green and blue color components tend to turn into red and yellow, respectively. This was associated with the increase in darkness. 

The variance in color among the irradiated NC films and the pristine film is known as color intensity (ΔE), which was estimated by means of the equation used in our previous work [27]. The correspondence of ΔE with the γ dose is displayed in Figure 10b. The values of ΔE increased while raising the γ dose up to 100 kGy. ΔE achieved permanent color difference since its values were greater than 5 [61,62]. This indicates that the PVA-Lg/Pd NC has a tendency towards color changes by γ irradiation. The changes in color are formed by the chemically active free radicals that were created by degradation. Furthermore, the chemically active free radicals that possess electrons with non-paired spin cause changes in color [27].

## 4. Conclusions

In the present work, PVA-Lg/Pd NC film was successfully prepared using ex-situ casting technique. The γ irradiations of PVA-Lg/Pd NC films cause degradation and chain crosslinking and consequently affect the thermal and optical properties. 

The XRD and DTA measurements show that the γ doses up to10 kGy enhance the thickness of lamellae, hence a rise in the melting temperature. Gamma doses 10–100 kGy generate defects that split the crystals, thus reducing the melting temperature. This can be due to crosslinking that increases the amorphous phase in the NC film, enhancing its resilience, thus the NC film can be a candidate for several industrial processes that require it to bend without breaking.

The TGA measurements signify the degradation of the NC films due to γ irradiation up to 10 kGy. This leads to the decomposition of the NC samples earlier than the pristine sample. At higher doses (10–100 kGy), crosslinking dominates. The domination of the crosslinking leads to an increase in the thermal stability of the NC films.

Both the refractive index and optical dielectric constant increased while raising the dose up to 100 kGy owing to crosslinking. This was linked to a reduction in the optical bandgap. The achieved optical changes may optimize PVA-Lg/Pd NC films for use in optoelectronic devices.

The pristine NC film displayed a distinctive response to color change by γ irradiation. The response in color change seemed obvious in the green and blue color constituents changing to red and yellow, linked to an increase in the darkness of the NC films.

## Figures and Tables

**Figure 1 polymers-14-02613-f001:**
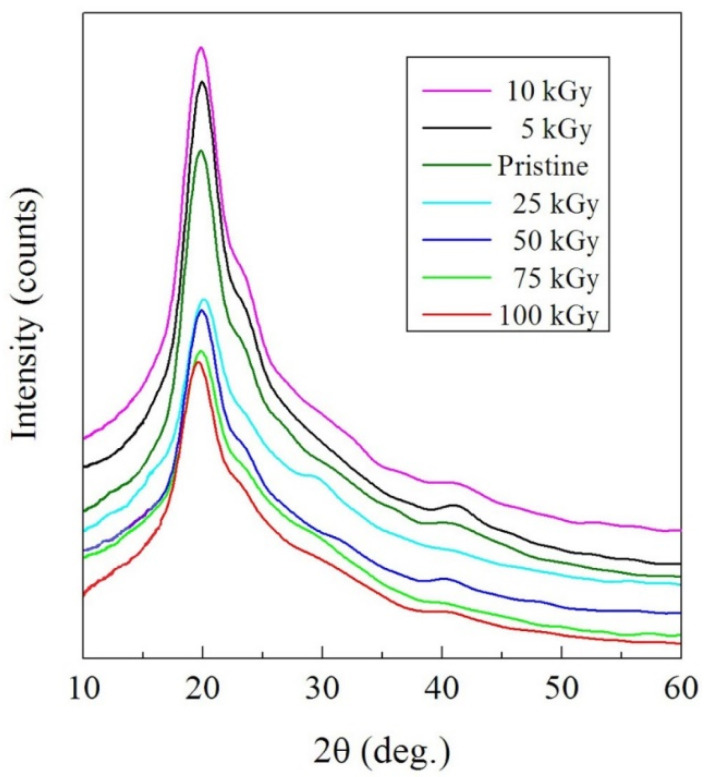
XRD patterns of the irradiated and pristine NC films.

**Figure 2 polymers-14-02613-f002:**
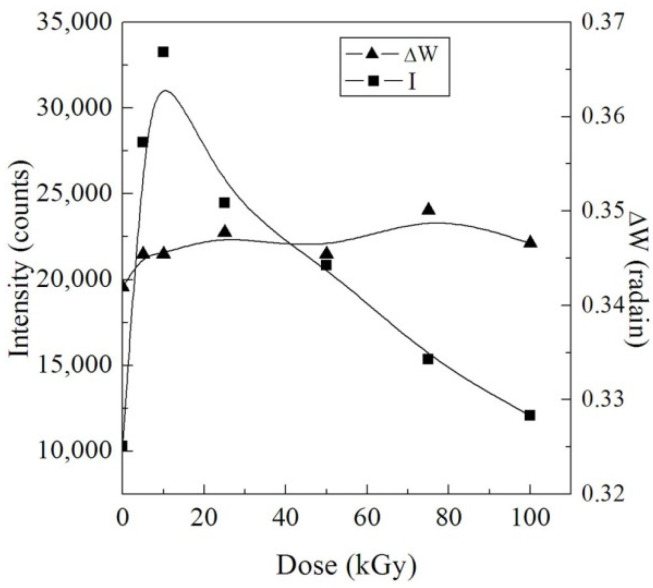
Variation of I and ΔW with the γ dose.

**Figure 3 polymers-14-02613-f003:**
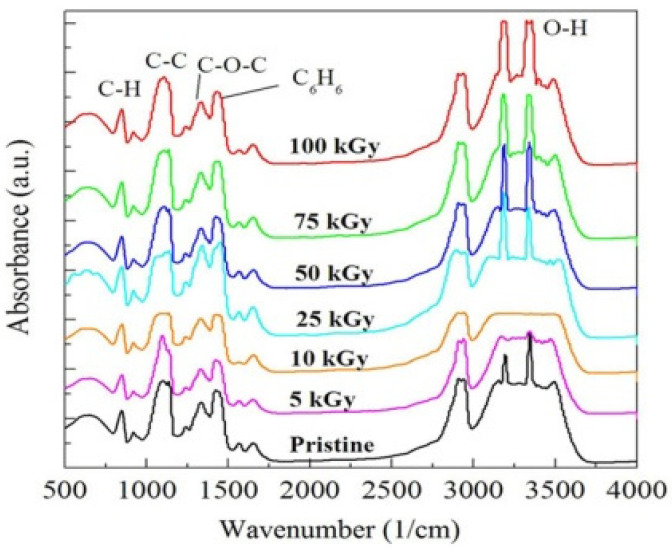
FTIR spectra of the irradiated and pristine NC films.

**Figure 4 polymers-14-02613-f004:**
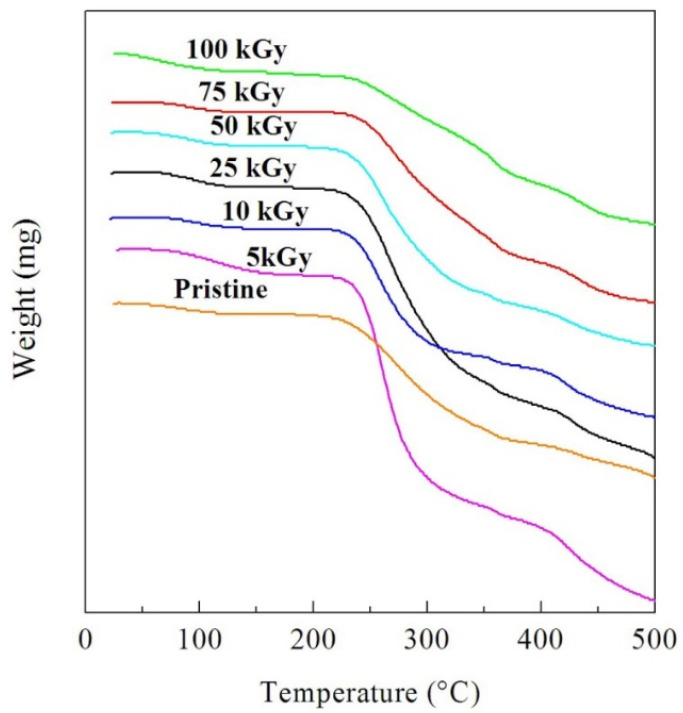
TGA curves of the pristine and γ irradiated NC films.

**Figure 5 polymers-14-02613-f005:**
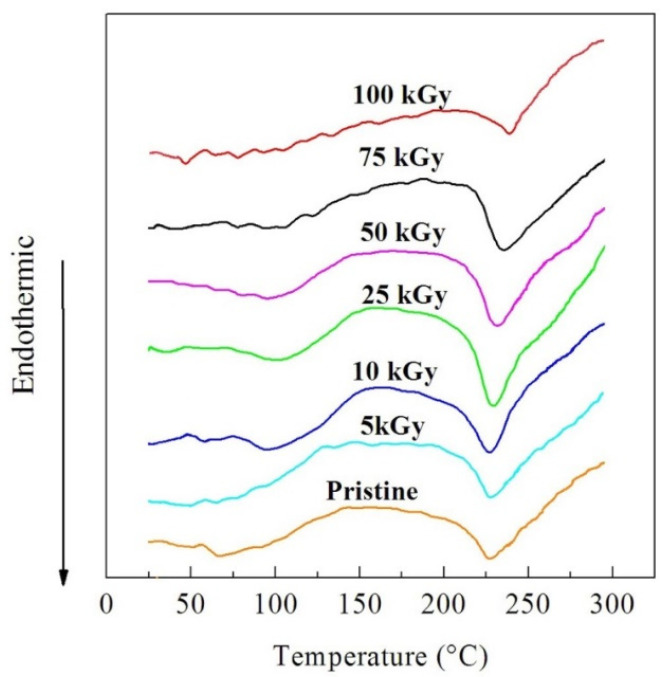
DTA thermograms of the pristine and irradiated NC films.

**Figure 6 polymers-14-02613-f006:**
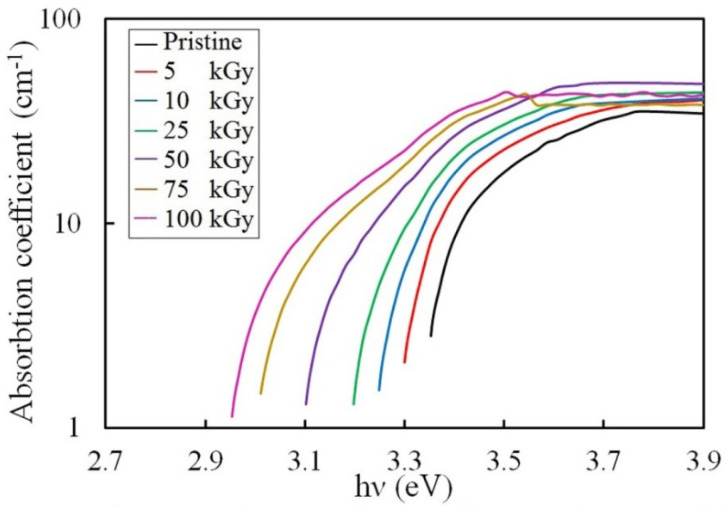
The absorption coefficient spectra of the irradiated and pristine NC films.

**Figure 7 polymers-14-02613-f007:**
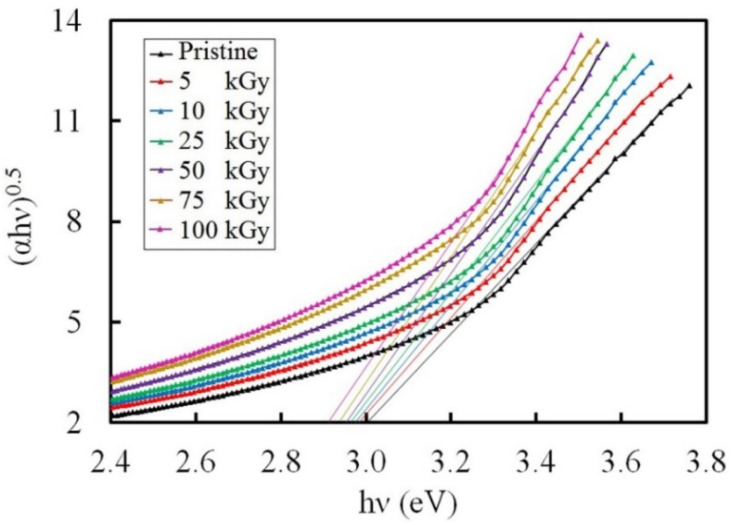
A Plot of (*α*hν)^0.5^ against hν for the pristine and irradiated NC films.

**Figure 8 polymers-14-02613-f008:**
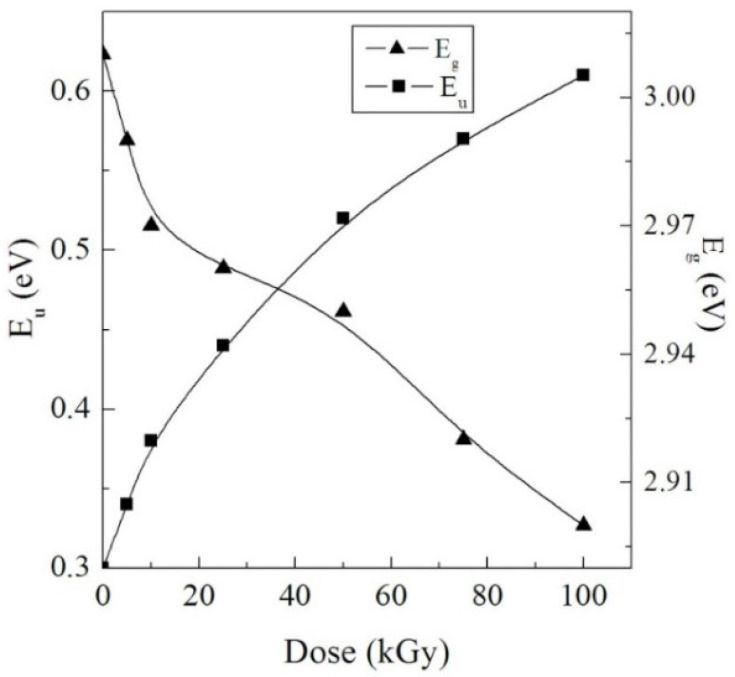
Variation of the E_u_ and E_g_ with the γ dose.

**Figure 9 polymers-14-02613-f009:**
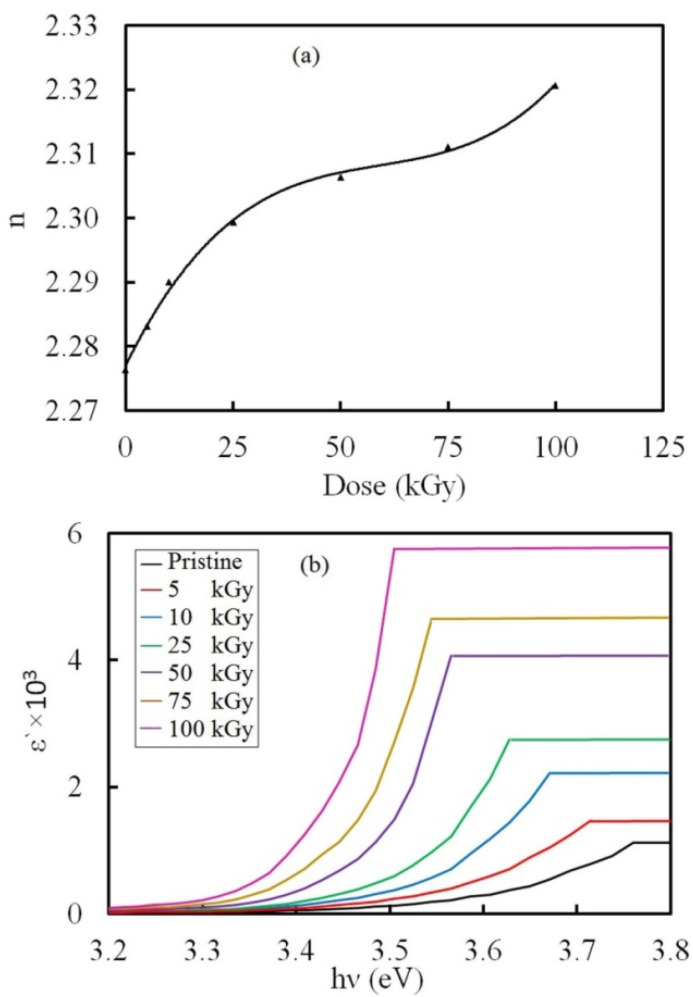
(**a**) Variation of n with the γ dose and (**b**) A plot of εʹ against hν for the pristine and irradiated NC samples.

**Figure 10 polymers-14-02613-f010:**
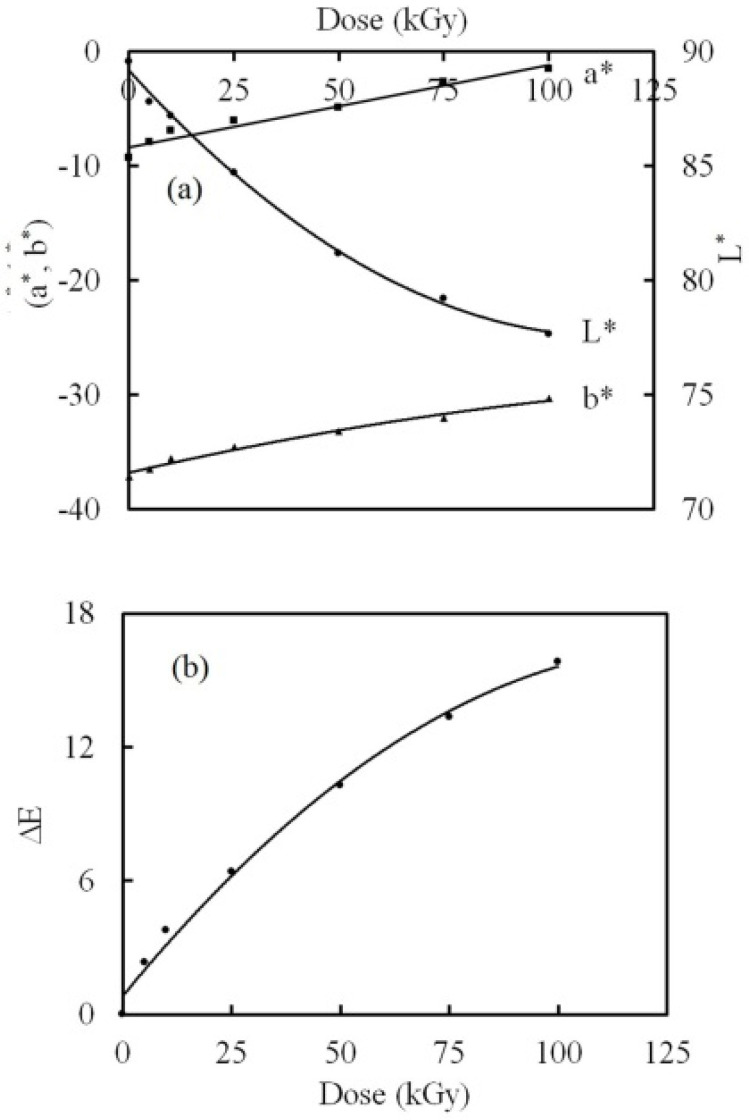
The dependence of the (**a**) color intercepts and (**b**) color intensity on the γ dose.

**Table 1 polymers-14-02613-t001:** Degradation temperatures (T_o_, T_1_), activation energies of thermal decomposition (E_a1_, E_a2_) of the two weight loss steps and melting temperature (T_m_) of the PVA-Lg/Pd NC films versus γ dose.

γ dose (kGy)	T_o_ (°C)	T_1_ (°C)	Ea_1_ (kJ/mole)	Ea_2_ (kJ/mole)	T_m_ (°C)
0	227	411	288	401	231
5	225	407	273	394	236
10	224	339	264	382	240
25	229	407	296	419	230
50	232	414	318	432	228
75	235	416	334	438	227
100	239	428	341	446	225

**Table 2 polymers-14-02613-t002:** The tristimulus values (X, Y, Z) and chromaticity coordinates (x, y, z) of the PVA-Lg/Pd NC samples against γ dose.

γ Dose (kGy)	X	Y	Z	x	y	z
0	82.78	75.36	42.59	0.381	0.362	0.247
5	76.66	70.58	41.80	0.384	0.372	0.244
10	73.14	67.93	40.61	0.393	0.378	0.233
25	69.82	65.39	39.54	0.400	0.384	0.226
50	63.70	60.62	37.67	0.403	0.389	0.224
75	57.04	55.23	36.19	0.406	0.394	0.221
100	53.85	52.60	34.85	0.412	0.397	0.212

## Data Availability

The data presented in this study are available on request from the corresponding author.

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
