# Peer review of "Structural, Thermal, and Optical Studies of Gamma Irradiated Polyvinyl Alcohol-, Lignosulfonate-, and Palladium Nanocomposite Film"

_polymers, 2022, doi:10.3390/polym14132613_

Round 1
Reviewer 1 Report
The manuscript: “Structural, thermal and optical studies of gamma irradiated PVA-Lg/Pd nanocomposite films” by F. Gharbi. et al., reports the results of synthesis and investigation of nanocomposite (NC) films of polyvinyl alcohol (PVA) with lignosulfonate (Lg) and nanosized palladium (Pd) NPs - PVA- Lg/Pd irradiated with 5-100 kGy γ doses. The effect radiation doses on the structural, thermal and optical properties of the NC films have been investigated using X-ray diffraction (XRD), Fourier transform infrared spectroscopy (FTIR), thermogravimetric analysis (TGA), differential thermal analysis (DTA) and UV–Vis spectroscopy. It was found that the irradiation with doses up to 23 - 100 kGy improves the degradation temperature from 227 to 239 C, indicating an increase in thermostability of the NC films, the melting temperature decreased from 240 to 225 °C meaning an increase of the amorphous phase. The authors attribute these modifications to the dominance of crosslinking that destroys crystallinity and improves the thermostability. The FTIR confirmed the domination of crosslinking that affected the optical character of the NC films. It was shown that the refractive index increased from 2.21 to 2.32 with increasing the dose up to 100 kGy. The obtained results reveal that the radiation enhances the spreading of Pd NPs in the blend matrix making more compact structure of the PVA-Lg/Pd NC.
The results presented by the authors are of interest for research in the field of irradiated polymers. In my opinion, the topic of this study is of general interest. At the same time, there are the following questions and recommendations for this manuscript.
- I recommend to more clearly formulate the purpose and novelty of this study in the introduction and conclusion, in order to show the contribution of the authors to the development of work in this area.
- The authors mentioned many methods, such as XRD, FTIR, TGA, DTA, however, in my opinion, for this type of research, the change in the resistivity of the samples after irradiation, measured by the four-wire probe method, is also very important. I recommend adding such measurements to the manuscript
- Figures 7 - 9 can be represented as fig. 7 a, b, c for a better understanding of these analysis results.
- I recommend to add a decoding of some abbreviations like "Lg" in the introduction for the convenience of readers.
- Finally, I recommend improving the style of this manuscript by reducing its length, as it looks a bit long and difficult to read.
In conclusion, I believe that the topic of this manuscript is consistent with the topic of Polymers. At the same time, the manuscript needs minor revision in accordance with the above recommendations. A revised manuscript may be resubmitted to Polymers.
Author Response
Really, mant thanks for your valuable comments
- I recommend to more clearly formulate the purpose and novelty of this study in the introduction and conclusion, in order to show the contribution of the authors to the development of work in this area.
Done and highlighted with green inside the text
- The authors mentioned many methods, such as XRD, FTIR, TGA, DTA, however, in my opinion, for this type of research, the change in the resistivity of the samples after irradiation, measured by the four-wire probe method, is also very important. I recommend adding such measurements to the manuscript
We believe that this will enhances the article, but unfortunately it is not possible for us to add it.
- Figures 7-9 can be represented as fig. 7 a, b, c for a better understanding of these analysis results.
Done
- I recommend to add a decoding of some abbreviations like "Lg" in the introduction for the convenience of readers.
Done
- Finally, I recommend improving the style of this manuscript by reducing its length, as it looks a bit long and difficult to read.
Done.
Reviewer 2 Report
Gharbi et al. prepared gamma irradiated PVA-Lg/Pd nanocomposite film. This manuscript need much more improvement in each section of the whole manuscript. Lack of logic and flow. Many texts are not clear. Typo errors inside. Plagrism is 43%. Reduce it less than 20%. Many sections directly copied.
Specific comments:
In the title what is PVA-Lg/Pd? write its full name for better understand to readers.
Line 24 write clearly oC? Check throughout the manuscript.
Authors need o highlighted importance of gamma irradiation of the composite film. What are demerits.
Line ex situ should have written as ex-situ. Abstract should be rewritten especially from line 23-28.
Line 86 -88, should be different paragraph and describe more details what authors did in this study.
Line 41, what is NCs and NPs, not defined.
Section 2.1 all the chemical country name, and other details need to mention.
Section 2.1.1 and 2.1.2 already authors perform their previous study no need to mention. Authors can just write we did as per previous study and cite your paper. Authors may put this information in supplementary information.
Section 2.2.2 more details in each technique need to mention.
XRD and FTIR not defined. check all abbreviations.
Section 3.1 meaning less to mention here.
What is results for PVA-Lg/Pd without gamma irradiation. Did authors compare. I don’t find anything.
Line 147, it should be mention in expt. Part. Here authors can why they do XRD. And what information XRD provide. Like this.
All the curves difficult to identify. Make it different color.
All the figure resolution should enhance.
Fig. 1 y –axis tik level should be removed just write intensity (a.u.).
Line 202, need references, insert it, Journal of Luminoscense, 228 (2020) 117593.
All the equation need to formatted as per text. Looks different. Check it carefully.
Line 225-229, need references. Carbohydrate Polymers 257 (2021) 117633.
All the bands of FTIR need to be summarized in a tabular form otherwise no one can understand.
Line 227-228, what decrease and increase, mention one particular phenomenon with wave numbers.
Fig. y a-axis no need to label.
The following references indicate the text accordingly https://doi.org/10.1007/s10924-022-02454-w, https://doi.org/10.1016/j.ijbiomac.2015.11.064, Polym. Eng. Sci., 62(5) 2022, 1526-1537: https://doi.org/10.1016/j.ijbiomac.2017.01.010,
XRD scans of the irradiated and pristine NC films., caption need to rewritten. Authors present XRD patterns not scan and authors present pure PVA and NC films.
All figure caption need to check and modified accordingly.
Line 252-253, no need to mention in results part, it should be in expt. Part.
Authors claimed crosslinking, did authors determined swelling or crosslinking density. please provide it.
In fig. 5. DTA have endothermic peak. There is no such phenomenon can be identifying in DTA.
Endo and exo. phenomenon can be analysis in DSC.
Fig. 7,8,9 and 10, make all into one figures with subfigures as a, b, c d..
Fig.11 ,12 make it into 1.
Fig. 13 and 14, into 1.
Authors claimed this is NCs but there is no evidence NCs. Please provide HRSEM or TEM of the samples. Also provided XPS data.
References part are disorder check it and correct it.
Correct English throughout the manuscript.
How NCs raw materials interact please provide a scheme.
Perspectives application need to mention in the manuscript.
Author Response
Many thanks for your valuable comments
In the title what is PVA-Lg/Pd? write its full name for better understand to readers.
Done and highlighted with yellow inside the text
Line 24 write clearly oC? Check throughout the manuscript.
Done
- Authors need o highlighted importance of gamma irradiation of the composite film. What are demerits?
Done and highlighted with yellow inside the text
Line ex situ should have written as ex-situ. Abstract should be rewritten especially from line 23-28.
Done and highlighted with yellow inside the text
Line 86 -88, should be different paragraph and describe more details what authors did in this study.
Done and highlighted with green inside the text
- Line 41, what is NCs and NPs, not defined.
Done
Section 2.1 all the chemical country name, and other details need to mention.
Done and highlighted with yellow inside the text
Section 2.1.1 and 2.1.2 already authors perform their previous study no need to mention. Authors can just write we did as per previous study and cite your paper. Authors may put this information in supplementary information.
Done
Section 2.2.2 more details in each technique need to mention.
XRD and FTIR not defined. check all abbreviations.
Done and highlighted with yellow inside the text
Section 3.1 meaning less to mention here.
Done
All the curves difficult to identify. Make it different color.
Done
Fig. 1 y –axis tik level should be removed just write intensity (a.u.).
Done
All the equation need to formatted as per text. Looks different. Check it carefully.
Done
Fig. y a-axis no need to label.
Done
XRD scans of the irradiated and pristine NC films., caption need to rewritten. Authors present XRD patterns not scan and authors present pure PVA and NC films.
Done
We think that no need to present pure PVA as the comparison is with the pristine NC film.
All figure caption need to check and modified accordingly.
Done
Authors claimed crosslinking, did authors determined swelling or crosslinking density. please provide it.
It is not possible for us
In fig. 5. DTA have endothermic peak. There is no such phenomenon can be identifying in DTA. Endo and exo. phenomenon can be analysis in DSC.
Yes, but in our study, endothermic peak only appeared.
Fig. 7,8,9 and 10, make all into one figures with subfigures as a, b, c d..
Done
Fig.11 ,12 make it into 1.
Done
Fig. 13 and 14, into 1.
Done
Authors claimed this is NCs but there is no evidence NCs. Please provide HRSEM or TEM of the samples. Also provided XPS data.
It is not possible for us
Round 2
Reviewer 2 Report
(Reject) Find the comment in the PDF file.

Author Response
Dear Sir
Really, many thanks for your kind comments; most of the comments were positive for us, so we performed it. But some comments, in my point of view, turned me disappointed when I read it, especially the following comments:
- What is results for PVA-Lg/Pd without gamma irradiation. Did authors compare. I don’t find anything.
The paper includes 14 figures and 3 tables. All figures contains the non-irradiated sample (whether spectrum or experimental point), similarly are the tables. All of the interpretation accounts on the effect of gamma radiation. What you mean by this comment? Why did you say I don’t find anything?
- Line 202, need references, insert it, Journal of Luminoscense, 228 (2020) 117593.
Line 225-229, need references. Carbohydrate Polymers 257 (2021) 117633.
The following references indicate the text accordingly https://doi.org/10.1007/s10924-022-02454-w, https://doi.org/10.1016/j.ijbiomac.2015.11.064, Polym. Eng. Sci., 62(5) 2022, 1526-1537: https://doi.org/10.1016/j.ijbiomac.2017.01.010,
I downloaded 2 of them and felt, in my point of view, that they will not add. I’m an associate editor of two significant journals and don’t encourage citation of papers recommended by reviewers that are not greatly related to work, in my point of view.
- For the description of FTIR results, we used to express the results in terms of figures and tables as you recommended, may be up to 2018, until a reviewer refused to put tabulated data based on the fact that there is no significant changes in the position of the peaks (wavenumbers). When searching, I found several publications up to 2022 show the FTIR results qualitatively, not quantitatively as in our case.
- In fig. 5. DTA have endothermic peak. There is no such phenomenon can be identifying in DTA. Endo and exo. phenomenon can be analysis in DSC.
I know well that the DTA or DSC thermograme should be characterized by the appearance of exothermic and endothermic peaks express the transition temperatures. In our results it is found that the DTA curve was characterized by the appearance of an endothermic peak at the melting temperature. This is not strange trend or it is the first time to get this trend. There are several publications that show that, whether for us or for others.
- You are right when you talked about the high percent of plagiarism and you mentioned that it should be less than 20 %. I used to send my papers with 5-10%. This time I faced problems in getting the software. So I contact the journal and arranged with them in treating it. It would be helpful from you if you sent to me the previous report.
Many thanks for your positive comments and kind cooperation.
Kind Regards
Round 3
Reviewer 2 Report
There is no improvement after 2nd revision. Still, there is 48% plagiarism. This is the author's responsibility to detect and remove it.
TEM, FESEM, and XPS are necessary for the nanocomposite film.
PVA/Pd nanocomposite already reported. I don't find the novelty of the study.
Authors always want to avoid reviewers' comments. In the first response, even authors avoid the comments in the response file.
Author Response
Many thanks for your kind cooperation. Kindly find our response to the comments:
- Still, there is 48% plagiarism. This is the author's responsibility to detect and remove it.
We treated the plagiarism. Now it is 22% Kindly find attached the iThinticate report. Many thanks
- TEM, FESEM, and XPS are necessary for the nanocomposite film.
Unfortunately it is not possible for us at current time. I’m very sorry and hope to accept our apologize. But maybe some of the results prove the dispersion of NPs in the blend matrix, such as “we didn’t observe any diffraction peak belonging to the NPs in the pattern of the NC film; signifying a full dispersion of the Pd NPs in the PVA-Lg matrix”.
- PVA/Pd nanocomposite already reported. I don't find the novelty of the study.
Our previous paper was on the effect of laser radiation on Pd-PVA NCP, while our present one is on the effect of gamma on PVA-Lg/Pd. The addition of Lg caused some medications on the Pd-PVA NCP. For example, the color intensity of Pd-PVA increased from 0 to 12. While after adding Lg, the color difference became greater (0-17) which makes the NC more suitable for several applications.
Many thanks again
